# Norepinephrine-Induced DNA Damage in Ovarian Cancer Cells

**DOI:** 10.3390/ijms21062250

**Published:** 2020-03-24

**Authors:** Rocio Lamboy-Caraballo, Carmen Ortiz-Sanchez, Arelis Acevedo-Santiago, Jaime Matta, Alvaro N.A. Monteiro, Guillermo N. Armaiz-Pena

**Affiliations:** 1Department of Basic Sciences, Division of Pharmacology, School of Medicine, Ponce Health Sciences University, Ponce, PR 00716, USA; rlamboy15@stu.psm.edu (R.L.-C.); jmatta@psm.edu (J.M.); 2Division of Cancer Biology, Ponce Research Institute, Ponce, PR 00716, USA; carmenortiz@psm.edu; 3Biomedical Sciences Program, University of Puerto Rico, Ponce, PR 00716, USA; arelis.acevedo2@upr.edu; 4Cancer Epidemiology Program, H. Lee Moffitt Cancer Center and Research Institute, Tampa, FL 33612, USA; alvaro.monteiro@moffitt.org; 5Division of Women’s Health, Ponce Research Institute, Ponce, PR 00716, USA

**Keywords:** norepinephrine, epinephrine, stress, adrenergic receptors, ovarian cancer, DNA lesions, cisplatin

## Abstract

Multiple studies have shown that psychological distress in epithelial ovarian cancer (EOC) patients is associated with worse quality of life and poor treatment adherence. This may influence chemotherapy response and prognosis. Moreover, although stress hormones can reduce cisplatin efficacy in EOC treatment, their effect on the integrity of DNA remains poorly understood. In this study, we investigated whether norepinephrine and epinephrine can induce DNA damage and modulate cisplatin-induced DNA damage in three EOC cell lines. Our data show that norepinephrine and epinephrine exposure led to increased nuclear γ-H2AX foci formation in EOC cells, a marker of double-strand DNA breaks. We further characterized norepinephrine-induced DNA damage by subjecting EOC cells to alkaline and neutral comet assays. Norepinephrine exposure caused DNA double-strand breaks, but not single-strand breaks. Interestingly, pre-treatment with propranolol abrogated norepinephrine-induced DNA damage indicating that its effects may be mediated by β-adrenergic receptors. Lastly, we determined the effects of norepinephrine on cisplatin-induced DNA damage. Our data suggest that norepinephrine reduced cisplatin-induced DNA damage in EOC cells and that this effect may be mediated independently of β-adrenergic receptors. Taken together, these results suggest that stress hormones can affect DNA integrity and modulate cisplatin resistance in EOC cells.

## 1. Introduction

Epithelial ovarian cancer (EOC) has one of the highest worldwide mortality rates among cancers of the female reproductive system [1]. The survival rate for women with EOC in the USA is generally low as evidenced by a five-year survival rate below 50% [2]. Poor survival rates and the limited efficacy of current therapies make uncovering novel molecular pathways involved in EOC a top priority.

Altered psychological states, such as chronic stress or depression, affect the quality of life of EOC patients and can also promote tumor progression and influence treatment outcomes [3,4,5,6,7,8,9]. Notably, a significant number of EOC patients lack social support, are clinically depressed, or exhibit anxiety disorders [4,10,11]. Moreover, several studies show that chronic stress correlates with elevated levels of systemic norepinephrine (NE) [12,13], increased EOC risk [14,15], and reduced chemotherapy efficacy [16,17,18]. In EOC, sustained adrenergic signaling is associated with increased inflammatory responses, tumor growth, invasiveness, and evasion of apoptotic processes [19,20,21,22,23,24,25,26,27].

In addition to the numerous immunologic effects of stress, recent evidence suggests that increased adrenergic stimulation can also promote cancer progression by compromising the genomic integrity of cancerous cells [28,29]. Different types of endogenous or exogenous events can cause DNA lesions which give rise to genetic mutations and can ultimately lead to genomic instability [29,30,31]. Dysregulated DNA repair processes can lead to the accumulation of DNA lesions and promote aberrant genomic events, such as gene mutations and chromosomal damage that leads to oncogenic transformation and tumor progression [30,32,33]. In vitro studies in breast cancer cell lines suggest that stress hormones are capable of inducing DNA damage by impairing DNA repair capacity and promoting cellular transformation [34]. In this study, we examined the ability of two catecholamines to induce single- or double-strand DNA damage, as well as the effect of NE in cisplatin-induced DNA damage in ovarian cancer cells.

## 2. Results

### 2.1. Dose-Dependent Norepinephrine (NE)-Induced DNA Double-Strand Breaks in EOC Cells

To explore whether NE induced single-strand or double-strand DNA damage in EOC cells in a dose-dependent manner, COV362 and SKOV3 cells were treated with increasing concentrations of NE (0.1, 1, 10, and 100 µM) for 24 h. As shown in Appendix A, NE (at 10 and 100 µM) significantly increased double-strand DNA damage in SKOV3 cells (Appendix A; *p* < 0.0001). In COV362 cells, NE only increased single-strand DNA breaks at 100 µM of NE (Appendix A; *p* < 0.05). Based on these data, we chose to expose EOC cells to 10 μM NE or epinephrine (EPI) for all remaining experiments.

### 2.2. NE and EPI Exposure Increases γ-H2AX foci Formation in Ovarian Cancer Cells

Immunofluorescence analyses for phospho-serine 139 histone H2AX (γ-H2AX) were performed in SKOV3, OV90, and COV362 cell lines exposed to NE (10 µM) or epinephrine (EPI) (10 µM) for either 1 h or 24 h in order to indirectly detect double-strand breaks (DSB). At both timepoints, NE- and EPI-treated cells had higher levels of γ-H2AX foci than untreated cells. This effect was observed in all three cell lines (Figure 1A–F). In addition, NE significantly induced more γ-H2AX foci formation than EPI at 1 h in COV362 cells (Figure 1E) and in all cell lines at 24 h (Figure 1A–F; *p* ≤ 0.05). These results show that these two stress hormones can induce double-strand DNA breaks in ovarian cancer cell lines at the experimental concentrations tested.

### 2.3. NE exposure Leads to Double, but not Single, Strand DNA Breaks, in Ovarian Cancer Cells

Our results show that NE exposure leads to higher levels of DNA damage than EPI. Thus, subsequent experiments focused on assessing NE-induced DNA damage. To confirm if NE induced DNA DSB in SKOV3, OV90, and COV362 cells we performed neutral comet assays. As shown in Figure 2A,B, SKOV3 cells exposed to NE for 1 h or 24 h had longer tails (comets) after electrophoresis indicating higher levels of DSB (*p* < 0.0001). The same effect was observed at both timepoints in OV90 (Figure 2C,D; *p* < 0.0001) and COV362 cells (Figure 2E,F; *p* < 0.0001).

To further characterize NE-induced DNA damage, DNA single-strand breaks (SSB) were assessed in SKOV3 and COV362 cells treated with NE (10 µM) for 1 h or 24 h by alkaline comet assays. In SKOV3 cells, NE exposure induced a slight increase in DNA damage at both timepoints studied (1 h and 24 h) when compared to untreated cells (Figure 3A). A similar effect was observed in COV362 cells (Figure 3B); however, this increase was not statistically significant (*p* > 0.05). Taken together, these data suggest that NE induces DSB, but not SSB.

### 2.4. β-adrenergic Receptor Blockade Abrogates NE-Induced DNA Damage

To elucidate whether the DNA damaging effect of NE is related to activation of β-adrenergic receptors, OV90, SKOV3, and COV362 cells were treated with 10 µM propranolol (non-selective β-blocker) for 30 min before exposure to NE for 1 h or 24 h. As shown in Figure 2A–F, NE-treated cells displayed increased levels of DSB (*p* < 0.0001). Furthermore, pre-treatment with propranolol prevented NE-induced DSB (*p* < 0.0001). Again, these effects were seen in all cell lines at both timepoints (Figure 2A–F). These data suggest that β-adrenergic receptor blockade may protect cells from NE-induced DSB DNA damage.

NE did not induce significant SSB DNA damage in SKOV3 cells after 1 h (Figure 3C). A similar effect was observed at 24 h (Figure 3D). In COV362 cells, NE exposure induced slight SSB DNA damage when compared to untreated cells (Figure 3E,F; not significant). At both timepoints, pre-treatment with propranolol blocked NE-induced SSB DNA damage (Figure 3E,F). Propranolol alone did not induce significant SSB or DSB DNA damage (Figure 3C–F). These results suggest that the subtle SSB DNA damaging effect of NE could be mediated by β-adrenergic receptors.

### 2.5. NE Stimulation Protects Ovarian Cancer Cells from Cisplatin-Induced DNA Damage

To assess the effect of NE on cisplatin-induced DSB in ovarian cancer cells, SKOV3 and COV362 cells were treated with cisplatin and/or NE for 24 h. In both cell lines, NE or cisplatin (SKOV3: 20 μM; COV362: 13.57 μM) treatment induced significant amounts of DSB DNA damage (Figure 4A,B; *p* < 0.0001). Surprisingly, cells treated with a combination of cisplatin and NE displayed significantly less DSB than cells treated with only cisplatin or NE (Figure 4A,B; *p* < 0.0001). These data suggest that NE is capable of inducing DSB while at the same time protecting cells from cisplatin-induced DNA damage.

Likewise, cisplatin treatment induced a significant increase in SSB in SKOV3 cells when compared with untreated cells at 24 h (Appendix A; *p* < 0.01). When cells were exposed to a combination of cisplatin and NE, a reduction in SSB DNA damage was observed when compared to cisplatin alone at the two timepoints studied. However, this effect was only significant at the 24-h timepoint (Appendix A; *p* < 0.05). In COV362 cells, cisplatin treatment increased SSB DNA damage almost two-fold when compared to untreated cells at 24 (*p* < 0.01) and 48 (*p* < 0.05) hours (Appendix A). Notably, both cell lines contain *TP53* mutations; however, COV362 also contains a mutation in *BRCA1*, which is also involved in several repair mechanisms and helps maintain genomic stability. Thus, the cytotoxic effect of cisplatin may vary between cell type due to their different genomic background [35] and their ability to respond to cisplatin-induced DNA adducts. Additionally, when cells were treated with a combination of NE and cisplatin, a reduction in SSB DNA damage was observed at both timepoints when compared to cisplatin-treated cells, but these changes were not statistically significant (Appendix A; not significant). Overall, these results show that NE exposure may alter the DNA-damaging response to cisplatin of ovarian cancer cells.

To determine whether NE-induced DNA damage reduction in cisplatin-treated cells was due to adrenergic activation, SKOV3 and COV362 cells were pre-treated with 10 µM propranolol for 30 min before adding cisplatin and/or NE treatment for 24 h (Figure 4 and Figure 5). As shown in Figure 4A,B, NE blocked DSB induced by cisplatin in both cell lines. On the other hand, propranolol pre-treatment followed by exposure to a combination of cisplatin and NE did not completely block NE-induced effects on cisplatin DSB in both cell lines (Figure 4A,B; not significant). Similar effects were observed for SSB in cells pre-treated with propranolol (10 µM) and then exposed to cisplatin plus NE (Figure 4C,D; not significant). Moreover, immunofluorescence analyses for γ-H2AX were performed to assess DSB after propranolol, NE, and cisplatin treatment for 24 h in SKOV3 cells. Similar to neutral comet assay analyses, NE and cisplatin treatment induced DSB, as shown by increased nuclear γ-H2AX foci formation (Figure 5; *p* < 0.01). Additionally, nuclear γ-H2AX foci were reduced in cells treated with a combination of cisplatin and NE (Figure 5). Lastly, these results demonstrate that pre-treatment with propranolol blocked NE-induced nuclear γ-H2AX foci formation (Figure 5; *p* < 0.01). However, propranolol did not completely abrogate NE-induced effects on cisplatin DSB (Figure 5). Taken together, these data suggest that NE can reduce cisplatin-induced DNA breaks. This effect can be partially mediated by adrenergic receptors or an alternative adrenergic-independent mechanism.

## 3. Discussion

The main focus of this study was to characterize the effect of NE and EPI exposure on DNA DSB and SSB in ovarian cancer cell lines. In normal individuals, basal circulating levels of NE range from 10–1000 pM and under stress conditions levels may increase up to 100 nM [36]. Elevated levels of NE and EPI are found in individuals with acute or chronic stress [36]. Additionally, due to direct sympathetic innervation [37,38] and local catecholamine biosynthesis [39,40], concentration of catecholamines (such as NE) can reach levels as high as 10 µM in the parenchyma of the ovary and the ovarian tumor microenvironment [37]. Thus, the main dose of NE used in our study (10 µM) was selected to mimic local conditions of these hormones at the tumor microenvironment. 

While each adrenergic receptor sub-type (α and β) leads to different physiologic functions, the effects of adrenergic signaling in EOC have been shown to be largely mediated by β-adrenergic receptors in tumor cells, specially stimulation β_2_-AR [6,14,18,21,22,25]. For this reason, our study evaluated the effects of β-AR blockage on NE-induced DNA damage. Furthermore, all three cell lines used in this study were confirmed to express β-1 and β-2 adrenergic receptors (data not shown). It should, however, be noted that EPI and NE have different binding affinities for each receptor subtype and that receptor heterogeneity can account for differences observed between cell lines (Appendix A) and differences in the response to EPI or NE (Figure 1).

EOC is mainly characterized by mutations in *TP53* or *BRCA1/2* [35,41,42,43]. p53 is a tumor suppressor protein that responds to DNA damage and exerts genome-guarding functions by mainly transactivating genes involved in cell cycle progression, DNA repair, and apoptosis [44,45]. In addition, the BRCA1/2 proteins play a crucial role in the repair of DSB mainly through the homologous recombination (HR) DNA repair pathway; however, *BRCA1* is also involved in many other pathways responsible for maintaining genomic stability [46]. 

The results of this study showed that NE induces DSB in all cell lines studied. Both NE and EPI increased γ-H2AX foci formation in all cell lines studied (*p* < 0.05). When a DSB occurs, the histone2AX complex rapidly becomes phosphorylated from two processes, namely activation of ATM following DSB or activation of ATR upon inhibition of DNA replication [47,48,49]. Consequently, large numbers of γ-H2AX molecules accumulate in the chromatin around the break site, creating a focus of this protein [48] and initiating the recruitment of other DNA damage response (DDR) molecules [50]. Currently, our group is working on determining which DDR kinase is the main trigger responsible for the observed γ-H2AX foci formation.

Previous studies have shown that catecholamines (specially NE) can have protective and DNA damaging effects in normal ovarian cells [51]. These hormones can act as antioxidants by scavenging free radicals and protecting DNA from reactive oxygen species (ROS) [52,53]. On the other hand, catecholamine oxidation reactions can result in the formation of unstable quinones and adrenochromes, which can cause DNA damage [54,55,56]. A study by Djelic et al. showed that NE exposure induced DNA damage in human lymphocytes, but this effect was reversed in the presence of the scavenging enzyme catalase [54]. In addition, a study performed in ovarian granulosa cells showed that 10 nM NE increased the levels of ROS without altering cellular viability, which was attributed to NE intracellular uptake by a membrane transporter for NE (NET) and acting independently from β-adrenergic receptor activation [57]. These findings suggest that one potential mechanism for adrenergic-mediated DNA damage might be enhanced ROS production.

Cisplatin-based therapy has been the most frequently used treatment for EOC for the last four decades [58]. Cisplatin’s main mechanism of action is related to its ability to promote cell death by binding to DNA and forming bulky cisplatin-DNA adducts that can block transcription leading to SSB or DSB and consequently overwhelming the cell’s response to DNA damage [59]. In addition, recent studies have shown that cisplatin generates oxidative stress, which also contributes to its anti-tumor effects. Cisplatin also enhances the formation of DNA SSB and DSB by hydrated electrons and hydroxyl radicals [60] and is a potent inducer of the nucleotide excision repair (NER) pathway.

A recent study by Yu et al. showed that only a small fraction of cisplatin (<10%) bound to the DNA will generate a significant increase in γ-H2AX foci formation [61]. This group also found that the main mechanism for these effects is the induction of oxidative stress, as observed by the ability of the free radical scavenger N-acetyl cysteine (a precursor of glutathione) to reverse cisplatin toxicity and γ-H2AX foci formation [61]. In addition, a recent study performed on non-tumorigenic, immortalized ovarian surface epithelial cells reported that 10 µM NE reduced the levels of DNA damage in the presence of ROS-generating molecules (i.e., bleomycin) [51]. These studies can help explain the results obtained in cisplatin and NE combined treatment experiments where NE treatment reduced the DNA damage induced by cisplatin. A possible explanation for this effect is the potential for NE to act as an antioxidant, thus reducing oxidative stress caused by cisplatin. The results obtained when combining propranolol, NE, and cisplatin provide further evidence to support this hypothesis. In order to validate this hypothesis, it is necessary to conduct additional experiments to measure ROS levels upon NE exposure, thereby evaluating the ability of NE to have a protective and damaging effect dependent on intracellular ROS levels.

Another possible explanation for the inhibitory effect of NE on cisplatin-induced DNA damage is that NE exposure leads to alterations in cell cycle progression (e.g., replication fork stalling). If NE inhibits replication, replication forks may not be able to pass through cisplatin-induced DNA adducts; therefore, DSB cannot be induced by cisplatin under NE exposure. This might explain why NE is able to activate DDR on its own and also prevent cisplatin-induced DDR activation. In order to assess this and determine a possible mechanism responsible for these results, we are currently working on investigating the effects on cell cycle progression after NE exposure and/or cisplatin treatment.

Taken together, these findings add to a growing body of literature on the role of stress hormones in cancer progression and the possibility that they may serve as an additional factor underlying the onset of chemoresistance. Our investigations into this area are still ongoing and we are working towards determining the mechanism behind the dual effect of NE on DNA integrity. On a larger scope, further cytogenetic studies (i.e., sister chromatid exchange and micronuclei formation) are needed to determine if NE-induced DNA damage is capable of producing stable genetic changes in EOC cells and provide further evidence of the deleterious effects of behavioral stress on cancer progression and treatment.

## 4. Materials and Methods 

### 4.1. Cell lines and Culture Conditions

SKOV3 and OV90 cells were maintained in RPMI 1640 supplemented with 10% fetal bovine serum and 1% antibiotic/antimycotic (Sigma, St. Louis, MO, USA, #A5955). Antibiotic/antimycotic is composed of penicillin (10,000 units), 10 mg streptomycin (10 mg), and amphotericin B (25 µg) per mL. COV362 were maintained in DMEM supplemented with 10% fetal bovine serum and 1% antibiotic/antimycotic. Cisplatin concentrations used for SKOV3 and COV362 cells were 20 µM and 13.57 µM, respectively. At these concentrations, cells tolerated 24 h of exposure to cisplatin while preserving more than 75% of cellular viability. All experiments were performed with cell cultures at no more than 80% confluency and cells at less than 20 passages. The study laboratory routinely screens for mycoplasma. COV362 cells were obtained from Sigma (7071910, Sigma, St. Louis, MO, USA), while SKOV3 and OV90 cells were obtained from the American Type Culture Collection (HTB-77 and CRL-11732, ATCC, USA). Cell line authentication was performed using short tandem repeat analysis.

### 4.2. Immunofluorescence, Imaging, and Image Analysis

Cells were plated in 4-well chamber slides (25,000 cells/well) and left to set overnight. The next day, cells were either left untreated (control) or treated once with of EPI (10 µM) or NE (10 µM). After time-point completion (1 h or 24 h), positive control cells were exposed to 30% H_2_O_2_ for 3 to 4 min, and then all samples were fixed with 4% formaldehyde for 10 min at 4 ºC. Cells were permeabilized with 5% PBS Triton X-100 for 10 min at room temperature, washed with cold PBS, and then blocked with 2% BSA in PBS at room temperature for 1 h. After blocking, cells were incubated with γ-H2AX (#2577, 1:400, Cell Signaling Technology, Denver MA, USA) overnight in a shaker at 4 ºC. After primary antibody incubation, cells were washed with cold PBS and incubated with AlexaFluor 488 (A11034, Invitrogen, Grand Island, NY, USA) at 1:600 dilution in PBS/2% BSA for 1 h at room temperature. Cells were washed several times with cold PBS and stained with 4′,6-diamidino-2-phenylindole (DAPI) (R37606, Invitrogen, Grand Island, NY, USA) for 5 min and mounted with Prolong Gold medium (P36934, Invitrogen, Grand Island, NY, USA). The same protocol was followed for the experiments with propranolol and the combination treatments of cisplatin and NE. For all experiments, slides were imaged using an Eclipse T_s_2r inverted microscope (Nikon Instruments, Mellvile, NY, USA) at 20× magnification. Fifty random cells were analyzed per condition by maxima quantitation using Image J software following the protocol described in the Duke University microscopy protocol guide [62].

### 4.3. Neutral Comet Assay

To assess DSB, SKOV3, OV90, and COV362, cell lines were plated in 6-well plates and then exposed to NE or left untreated for 1 h or 24 h the following day. A group of cells were exposed to 30% H_2_O_2_ for five minutes and used as positive controls. In all the experimental samples, the cell viability in Trypan blue exclusion test was acceptable (over 75%). Next, the neutral version of the CometAssay^®^ was performed per manufacturer’s instructions (Appendix A, Trevigen, Gaithersburg, MD, USA). Without alkaline treatment, this version of the neutral comet assay mainly detects DBS. Additionally, in order to reduce known causes of assay variability, all experiments were run using the Trevigen’s CometAssay^®^ Electrophoresis System. Next, to assess whether DNA damage was related to adrenergic receptor activation, cells were pre-treated for 30 min with propranolol (10 µM) (non-specific β-receptor antagonist) before hormone treatment. Importantly, for every experiment, cell viability was measured using trypan blue exclusion to ensure 75% or more viability per sample. Lastly, DNA was stained with SYBR Green (Trevigen, Gaithersburg, MD, USA), and slides were imaged using Eclipse T_s_2r inverted microscope (Nikon Instruments, Mellvile, NY, USA) at 20× magnification. Fifty random cells were selected per sample and analyzed for tail moment using OpenComet (open-source automated software) [63].

### 4.4. Alkaline Comet Assay

To assess SSB, SKOV3, and COV362, cells were exposed to NE at the timepoints required for each experiment: 1 h or 24 h. After NE exposure, the cells were collected and processed to perform the alkaline version of the CometAssay^®^ as described by the manufacturer (Trevigen, Gaithersburg, MD, USA). The nuclei were stained with Yoyo-1^®^ and visualized using the EVOS™ M7000 Imaging System (Thermo Scientific, Rockford, IL, USA). Fifty nuclei were randomly selected and analyzed for percentage of DNA in tail using the Comet Analysis Software (Trevigen, Gaithersburg, MD, USA). Ethyl methanesulfonate (EMS) (Sigma, St. Louis, MO, USA) was used as a positive control at a concentration of 12 mM for 4 h.

## Figures and Tables

**Figure 1 ijms-21-02250-f001:**
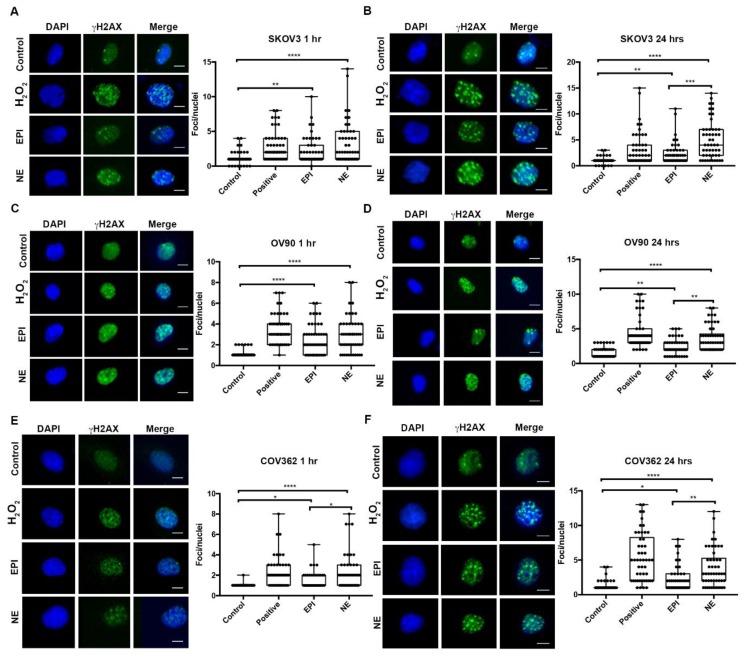
γ-H2AX foci formation after NE or EPI treatment in EOC cells. SKOV3 (**A**,**B**), OV90 (**C**,**D**), and COV362 (**E**,**F**) were treated with NE or EPI (10 µM), left untreated (control), or treated with 30% H_2_O_2_ (positive control) for 1 h or 24 h. After catecholamine exposure, cells were fixed and immune stained for γ-H2AX, and foci formation was assessed for fifty (50) cells per treatment. Statistical significance was determined by ANOVA with Dunn’s multiple comparison correction. (* *p* < 0.05, ** *p* < 0.01, *** *p* <0.001, **** *p* < 0.0001). Scale bar = 10 μM. Images are representative of three independent experiments.

**Figure 2 ijms-21-02250-f002:**
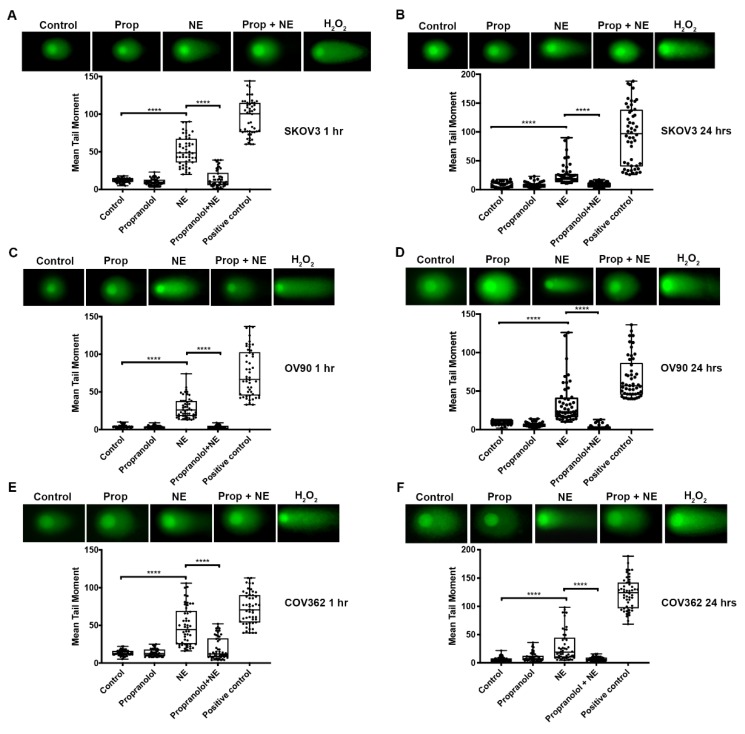
Propranolol abrogates NE-induced DNA double strand breaks. Neutral comet assays shown after exposure of SKOV3 (**A**,**B**), OV90 (**C**,**D**), or COV362 (**E**,**F**) cells to propranolol followed by NE (10 µM) for 1 h or 24 h. DNA double-strand breaks were measured using the mean tail moment. Fifty (50) comets were evaluated for each treatment. Data presented is representative of three independent experiments performed in duplicate. Statistical analysis was performed using one-way ANOVA with Dunn’s multiple comparison correction. (**** *p* < 0.0001).

**Figure 3 ijms-21-02250-f003:**
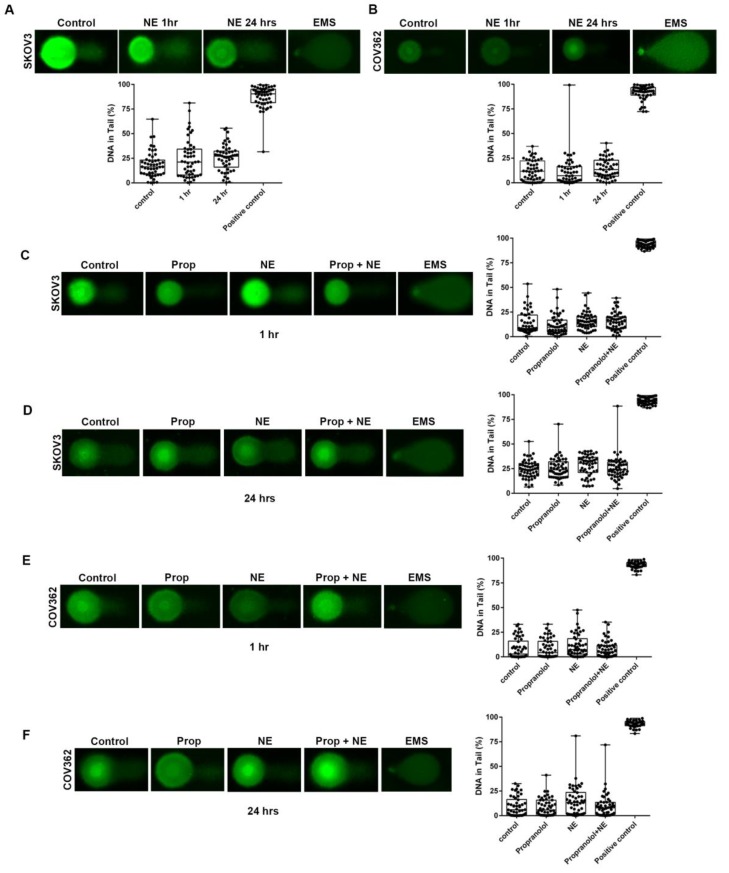
NE does not induce DNA single strand breaks in ovarian cancer cells. Alkaline comet assay after exposure NE (10 µM) in SKOV3 (**A**) and COV362 (**B**) cells after 1 h or 24 h. Alkaline comet assay after propranolol pre-treatment followed by NE exposure for 1 h or 24 h in SKOV3 (**C**,**D**) and COV362 cells (**E**,**F**). Single-strand breaks were measured using the percentage of DNA in tail. Fifty (50) comets were evaluated for each treatment. Data presented is representative of three independent experiments performed in duplicate. Statistical analysis was performed using one-way ANOVA with Dunn’s multiple comparison correction. No significant differences were observed among treatments (*p* > 0.05).

**Figure 4 ijms-21-02250-f004:**
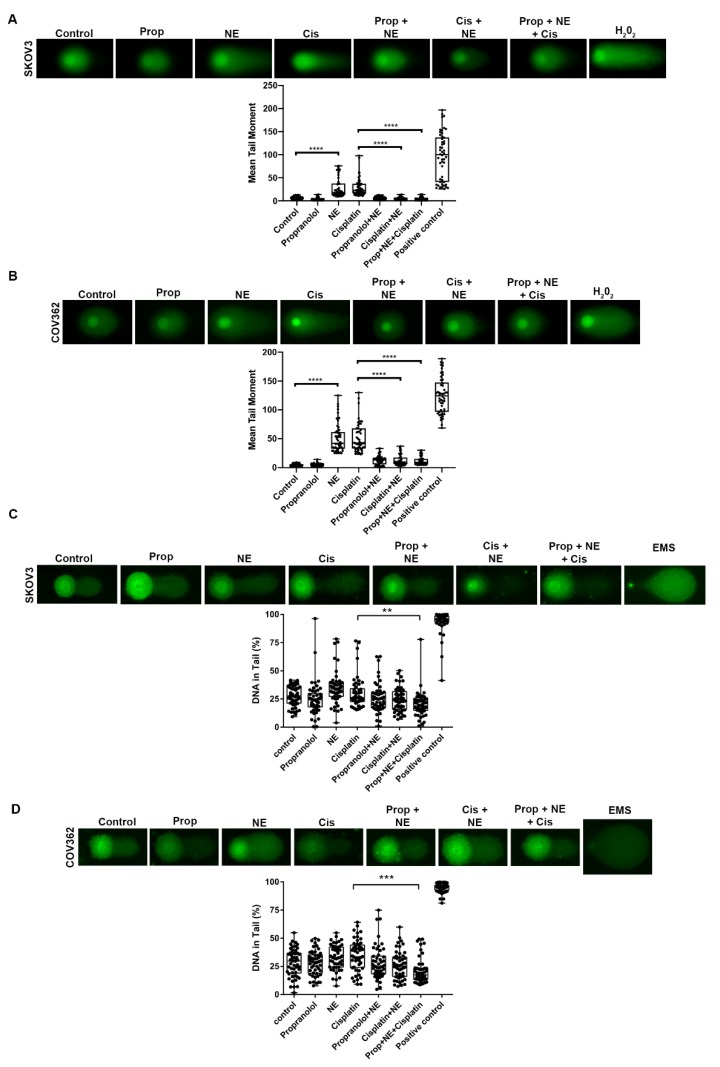
DNA single and double strand breaks after exposure of combination of propranolol, NE, and cisplatin in SKOV3 and COV362 cells. Neutral (**A**,**B**) and alkaline comet assay (**C**,**D**) after individual treatment with either NE, cisplatin, or propranolol, and after pre-treatment with propranolol followed by treatment of different agent combinations, for 24 h in SKOV3 and COV362 cells. Double-strand breaks were measured using mean tail moment, and single-strand breaks were measured using the percentage of DNA in tail. Statistical analysis was performed using one-way ANOVA with Dunn’s multiple comparison correction (** *p* < 0.01, *** *p* < 0.001, **** *p* < 0.0001). Data presented is representative of three independent experiments performed in duplicate.

**Figure 5 ijms-21-02250-f005:**
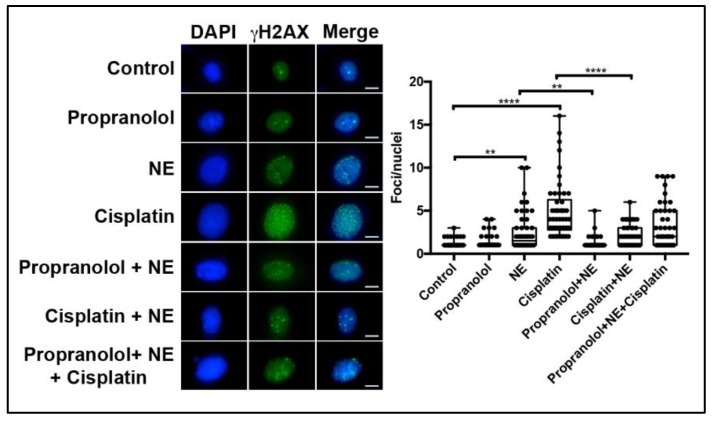
γ-H2AX foci formation after exposure of combination of propranolol, NE, and cisplatin in SKOV3 cells. Immunofluorescence analysis of SKOV3 cells after individual treatment with either NE, cisplatin, or propranolol, and after pre-treatment with propranolol followed by treatment of different agent combinations or left untreated (control) for 24 h. After timepoint completion, cells were fixed and immune stained for γ-H2AX and foci formation was assessed for fifty (50) cells per treatment. Statistical significance was determined by ANOVA with Dunn’s multiple comparison correction. (* *p* < 0.05, ** *p* < 0.01, **** *p* < 0.0001). Scale bar = 10 μM. Images are representative of three independent experiments.

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
