# Peer review of "Norepinephrine-Induced DNA Damage in Ovarian Cancer Cells"

_ijms, 2020, doi:10.3390/ijms21062250_

Round 1
Reviewer 1 Report
Comments are included as a pdf.

Reviewer 2 Report
Journal: International Journal of Molecular Sciences
Manuscript ID: ijms-745090
Type of manuscript: Article
Title: Norepinephrine-induced DNA damage in ovarian cancer cells
Authors: Guillermo Armaiz-Pena*, Rocio Lamboy-Caraballo, Carmen Ortiz-Sanchez, Arelis Acevedo-Santiago, Jaime Matta, Alvaro N.A. Monteiro
This manuscript has been described on the norepinephrine-induced DNA damage in ovarian cancer cells. The analyses of DNA damage were measured alkaline and neutral comet assays. As a result, norepinephrine reduced cisplatin-induced DNA damage in ovarian cancer cells. Furthermore, these results suggested that the stress hormones can induce DNA damage and promote cisplatin resistance. However, this manuscript has contained a few indistinct points. Therefore, it deserves to be published after major revision.
Major
1. Figures: The graphs of figures in manuscript are too small to read and evaluate. Authors should show the larger graphs in figures.
2. Figure 1, COV362 1 hr: The error bar of control in COV362 1 hr is very small. Are error bars described in COV362 1 hr?
Minor
3. Supplementary Figure 1 and Figure 2: Supplementary Figure 1 and Supplementary Figure 2 are indicated in authors’ manuscript. I think the supplementary figure will be shown on Web, not in manuscript.
4. Line 91-98: Not centering. Justification.
Round 2
Reviewer 1 Report
No comments necessary.
Reviewer 2 Report
Journal: International Journal of Molecular Sciences
Manuscript ID: ijms-745090
Type of manuscript: Article
Title: Norepinephrine-induced DNA damage in ovarian cancer cells
Authors: Guillermo Armaiz-Pena*, Rocio Lamboy-Caraballo, Carmen Ortiz-Sanchez, Arelis Acevedo-Santiago, Jaime Matta, Alvaro N.A. Monteiro
This manuscript has been described on the norepinephrine-induced DNA damage in ovarian cancer cells. The analyses of DNA damage were measured alkaline and neutral comet assays. As a result, norepinephrine reduced cisplatin-induced DNA damage in ovarian cancer cells. Additionally, these results suggested that the stress hormones can induce DNA damage and promote cisplatin resistance. Furthermore, reviewer received the appropriate answer from authors. Therefore, it deserves to be published in International Journal of Molecular Sciences.